# Bayesian Outcome Weighted Learning

**Sophia Yazzourh**
Institut de Mathématiques de Toulouse
UMR5219 - Université de Toulouse
CNRS - UPS IMT
F-31062 Toulouse Cedex 9, France
`sophia.yazzourh@math.univ-toulouse.fr`

**Nikki L. B. Freeman**
Department of Biostatistics and Bioinformatics
Duke Clinical Research Institute
300 W. Morgan Street
Durham, North Carolina 27701
`nikki.freeman@duke.edu`

## Abstract

A primary goal of statistical precision medicine is to learn optimal individualized treatment rules (ITRs). Outcome weighted learning (OWL) introduced a classification-based approach to this task. Here, we introduce the first Bayesian formulation of OWL. Starting from the OWL objective function, we generate a pseudo-likelihood which can be expressed as the sum of a scale mixture of normal distributions. Gibbs sampling is used to sample the posterior distribution of the parameters. In addition to providing a strategy for learning an optimal ITR, Bayesian OWL provides a natural, probabilistic approach to estimate uncertainty in ITR treatment recommendations themselves.

## 1 Introduction

Statistical precision medicine uses data to match patients to treatments to improve health outcomes [Kosorok and Laber, 2019]. This can be achieved through individualized treatment regimes (ITRs), functions mapping patient characteristics to treatment recommendations. One goal is to learn optimal ITRs that, when implemented, result in better outcomes on average in the target population compared to other strategies, such as a one-size-fits-all approach [Kosorok and Laber, 2019, Murphy, 2003].

Machine learning-based approaches for optimal ITR learning, such as outcome weighted learning (OWL) introduced by Zhao et al. [2012], convert the statistical ITR-learning problem into a classification problem, bypassing the need to estimate ancillary functions like the conditional mean of the outcome and uncoupling the quality of the estimated ITR from the quality of the ancillary function estimates. In contrast, OWL enables direct learning the optimal ITR. Because of its flexibility, many extensions to OWL have been developed [Zhao et al., 2015, Zhou et al., 2017, Liu et al., 2018, Fu et al., 2019, Zhao et al., 2019].

Bayesian approaches for learning optimal ITRs have also been proposed, including Bayesian machine learning (BML) [Murray et al., 2018] which employs a variant of approximate dynamic programming and likelihood-based methods that model both the distribution of final and intermediate outcomes within the Bayesian framework [Thall et al., 2002, 2007, Arjas and Saarela, 2010, Zajonc, 2012, Xu et al., 2016, Yu and Bondell, 2023].

Here we focus on the classification-based perspective for learning optimal ITRs. While these approaches are powerful in terms of predictive power, they do not have a natural mechanism for inference or for quantifying the uncertainty in the learned ITR's treatment recommendations. Yet capturing this kind of uncertainty is crucial for healthcare decision-making. To address this, we introduce Bayesian outcome weighted learning (Bayesian OWL), the first optimal ITR-learning method to directly learn optimal ITRs from the classification perspective. By transforming the OWL

Workshop on Bayesian Decision-making and Uncertainty, 38th Conference on Neural Information Processing Systems (NeurIPS 2024).

framework into a probabilistic model, we generate a posterior distribution that enables inference and uncertainty quantification for the treatment recommendations themselves.

## 2 Background

### 2.1 Setting

We let $A \in \mathcal{A} = \{-1, 1\}$ denote the action, or treatment, and assume that observed treatments are assigned randomly as in a clinical trial with $P(A = 1) = \rho$ known. Let $X_i = (X_{i,1}, \ldots, X_{i,p})^\top \in \mathcal{X}$ denote the p-dimensional biomarker and prognostic information vector, and let $R$ denote the outcome (bigger is better). We further assume that the reward can be rescaled so that $R > 0$. Then, the observed data is iid replicates of $(A_i, X_i, R_i)$ for $i = 1, \ldots, n$.

An ITR is a function $d$ that maps from patient features $\mathcal{X}$ to a recommended treatment in $\mathcal{A}$. For a given ITR $d$, the value of $d$ is $V(d) = \mathbb{E}[R(d)]$, where $R(d)$ is the reward we would observe if treatments were allocated according to rule $d$. An optimal ITR $d^{\text{opt}}$ satisfies $V(d^{\text{opt}}) \geq V(d)$ for all $d \in \mathcal{D}$, where $\mathcal{D}$ is a class of ITRs. Our goal is to learn an optimal ITR $d^{\text{opt}}$. Under the assumptions of causal consistency, the stable unit treatment value assumption, no unmeasured confounding, and positivity, $V(d)$ can be identified from the observed data and $V(d) = \mathbb{E}_\mathcal{X}\{\mathbb{E}[R|A = d(\mathbf{x}), X = \mathbf{x}]\}$.

### 2.2 Outcome weighted learning

If we let $P$ denote the distribution of $(X, A, R)$, and $P^d$ denote the distribution of $(X, A, R)$ when $A = d(X)$, then the reward we would expect if ITR $d(X)$ were followed is given by

$$\mathbb{E}^d(R) = \int R dP^d = \int R \frac{dP^d}{dP} dP = \mathbb{E}\left[\frac{\mathbb{1}(A = d(X))}{A\rho + (1 - A)/2} R\right]. \tag{1}$$

Zhao et al. [2012] showed that maximizing Equation (1) is equivalent to minimizing a weighted classification problem with the 0-1 loss function. By replacing the 0-1 loss with a confex surrogate loss, they formulated an objective function that can be efficiently minimized using machine learning techniques. To learn optimal ITRs, they proposed the OWL which minimizes

$$Q_n^{\text{OWL}}(\boldsymbol{\beta}) = \frac{1}{n} \sum_{i=1}^n \frac{r_i}{a_i\rho + (1 - a_i)/2} (1 -_i h(\mathbf{x}_i, \boldsymbol{\beta}))_+ \tag{2}$$

where the empirical measure replaces the true measure, $(z)_+ = max(z, 0)$ denotes the hinge loss function, and $h(\cdot)$ is the ITR parameterized by $\boldsymbol{\beta}$. Song et al. [2015] introduced a penalized variant of OWL that included a regularization term $p_\lambda(\boldsymbol{\beta})$ for the ITR parameters. POWL minimizes the objective function

$$Q_n^{\text{POWL}}(\boldsymbol{\beta}) \frac{1}{n} \sum_{i=1}^n \frac{r_i}{a_i\rho + (1 - a_i)/2} (1 - a_i h(\mathbf{x}_i, \boldsymbol{\beta}))_+ + \sum_{j=1}^p p_\lambda(|\beta_j|) \tag{3}$$

where $p_\lambda(\boldsymbol{\beta})$ is a penalty function and $\lambda$ is a tuning parameter.

### 2.3 Bayesian support vector machines

Although the pure machine learning framework is powerful, it is limited in its ability to capture and model uncertainty as in a statistical framework. Polson and Scott [2011] bridged this gap between pure machine learning and statistical modeling for SVMs by showing how to cast SVM into a Bayesian framework. They considered the $L^\alpha$-norm regularized support vector classifier that chooses $\beta$ to minimize

$$d_\alpha(\boldsymbol{\beta}, \nu) = \sum_{i=1}^n \max(1 - r_i \mathbf{x}_i^\top \beta, 0) + \nu^{-\alpha} \sum_{j=1}^p |\beta_j/\sigma_j|^\alpha \tag{4}$$

where $\sigma_j$ is the standard deviation of the $j$-th element of $\mathbf{x}$ and $\nu$ is a tuning parameter. For this objective function, the learned classifier is a linear classifier. Polson and Scott [2011] shows

that minimizing Equation (4) is equivalent to finding the mode of the pseudo-posterior distribution $p(\boldsymbol{\beta}|\nu, \alpha, r) \propto \exp(-d_\alpha(\boldsymbol{\beta}, \nu)) \propto C_\alpha(\nu)L(r|\boldsymbol{\beta})p(\boldsymbol{\beta}|\nu, \alpha)$, where $C_\alpha$ is a pseudo-posterior normalization constant. The main theoretical result from Polson and Scott [2011] is that the pseudo-likelihood contribution $L_i(r_i|\boldsymbol{\beta})$ is a location-scale mixture of normals (Polson and Scott [2011], Theorem 1).

## 3 Our approach

We follow the strategy employed by Polson and Scott [2011] to cast the OWL objective function into a probabilistic Bayesian learning framework. The conversion is not one-to-one since Polson and Scott [2011] constructed a Bayesian model for a standard SVM whereas the objective function for OWL Equation (2) is a weighted SVM problem. Throughout, we will assume $R > 0$. When this is not the case, a distance-preserving transformation of $R$ from $\mathbb{R}$ to $\mathbb{R}^+$ can be used. Assuming that $h$ is linear, i.e., $h(\mathbf{x}_i, \boldsymbol{\beta}) = \mathbf{x}_i^\top \boldsymbol{\beta}$ and following the strategy taken in Theorem 1 of Polson and Scott [2011], the contribution of a single observation to the pseudo-likelihood is given by

$$
\begin{aligned}
L_i(a_i|r_i, \mathbf{x}_i, \boldsymbol{\beta}) &= \exp\left\{-2\frac{r_i}{a_i\rho + (1-a_i)/2}\max(1 - a_i\mathbf{x}_i^\top\boldsymbol{\beta}, 0)\right\} \\
&= \sum_{\{k=1,-1\}} \mathbb{1}(a_i = k)\int_0^\infty \frac{1}{\sqrt{2\pi\lambda_i}}\exp\left\{-\frac{1}{2\lambda_i}\left(\frac{r_i}{\rho} + \lambda_i - \frac{r_i}{\rho}a_i\mathbf{x}_i^\top\boldsymbol{\beta}\right)^2\right\}d\lambda_i
\end{aligned}
$$
(5)

or in other words that $L_i(a_i, \lambda_i|r_i, \mathbf{x}_i, \boldsymbol{\beta})$ is a scale mixture of Gaussians.

### 3.1 Prior specification for the ITR parameters

In their formulation of Bayesian SVM, Polson and Scott [2011] use the exponential power prior for $\boldsymbol{\beta}$, a prior that can be show to be equivalent to L1-regularization of the regression parameters. Regularization of the OWL parameters has been explored as in Song et al. [2015]. In this paper, we first construct our method as an analogy to the original formulation of OWL without penalization. We make this choice because (1) our primary aim is to develop a Bayesian classification-based ITR learning approach, and because (2) L1-regularization does not necessarily yield sparse rules (see the discussion in Section 4.1 of Polson and Scott [2011]). However, regularization helps avoid overfitting, a common problem in machine learning. Thus, we also explore penalty priors for $\boldsymbol{\beta}$, including the exponential power prior distribution and the spike-and-slab prior distribution Table 1.

Table 1: Priors for $\boldsymbol{\beta}$

| Prior for $\boldsymbol{\beta}$ | $p(\boldsymbol{\beta})$ |
|---|---|
| Normal distribution | $\prod_{j=1}^p \frac{1}{\sqrt{2\pi\sigma_0^2}}\exp\left\{-\frac{1}{2}\frac{(\beta_j - \mu_{0,j})^2}{\sigma_0^2}\right\}$, where $\mu_0$ and $\sigma_0^2$ are hyperparameters of the Normal distribution |
| Exponential power distribution | $\prod_{j=1}^p \omega_j^{-\frac{1}{2}} \cdot \exp\left\{-\frac{1}{2\nu^2}\sum_{j=1}^p \frac{\beta_j^2}{\sigma_j^2\omega_j}\right\} \cdot \prod_{j=1}^p p(\omega_j|\alpha)$, where $\boldsymbol{\omega}$ and $\alpha$ are hyperparameters and when $\alpha = 1$ $p(\omega_j|\alpha) \sim Exponential(2)$ |
| Spike-and-slab distribution | $\prod_{j=1}^p \left[(\gamma_j N(0, \nu^2\sigma_j^2) + (1-\gamma_j)\delta_0(\beta_j))\pi^{\gamma_j}(1-\pi)^{1-\gamma_j}\right]$ where $\pi$ and $\gamma_j$ are hyperparameters and $\delta_0$ is the Dirac measure |

### 3.2 Estimation

To draw from the pseudo-posterior distribution, Polson and Scott [2011] employed two algorithms, an expectation-minimization (EM) approach and a Gibbs sampling approach. The approach we take is the latter. Although sampling the pseudo-posterior is likely to be more time intensive than estimation via the EM algorithm, the rationale for a fully Bayesian approach is to enable uncertainty

quantification (Section 3.3). Because of conjugacy, derivation of the Gibbs sampling algorithms is straightforward and is not presented here (see Yazzourh and Freeman [2024] for details).

### 3.3 Prediction and uncertainty quantification

Using the posterior predictive distribution, we can make treatment recommendations for a new patient and quantify our uncertainty in our recommendation. Let $\Theta = \{\boldsymbol{\beta}, \boldsymbol{\lambda}\}$ and $\tilde{a}$ denote the recommended treatment for a new patient with features $\tilde{\mathbf{x}}$. Then

$$p(\tilde{a} = 1 | \tilde{\mathbf{x}}, \mathbf{X}, \mathbf{r}, \mathbf{a}) = \int_{\Theta} p(\tilde{a} = 1 | \tilde{\mathbf{x}}, \mathbf{X}, \mathbf{r}, \mathbf{a}, \boldsymbol{\beta}, \boldsymbol{\lambda}) p(\boldsymbol{\beta}, \boldsymbol{\lambda} | \mathbf{x}, \mathbf{r}, \mathbf{a}) d\theta.$$

For the class predictions, we can use the probit model which has the form $p(a = 1 | x) = \Phi(x^\top \beta)$ where $\Phi$ is the cumulative distribution function of the standard normal distribution. Thus we can write the posterior predictive distribution as

$$p(\tilde{a} = 1 | \tilde{\mathbf{x}}, \mathbf{X}, \mathbf{r}, \mathbf{a}) = \int_{\Theta} \Phi(\tilde{\mathbf{x}}^\top \boldsymbol{\beta}) p(\boldsymbol{\beta}, \boldsymbol{\lambda} | \mathbf{x}, \mathbf{r}, \mathbf{a}) d\theta. \tag{6}$$

## 4 Simulation study

### 4.1 Classification performance

We conducted simulation studies to assess the classification performance of the proposed method, following Zhao et al. [2012] and Song et al. [2015]. We compared the performance of OWL, Bayesian OWL with normal priors for $\boldsymbol{\beta}$, Bayesian OWL with exponential power prior for $\boldsymbol{\beta}$, and Bayesian OWL with spike-and-slab prior for $\boldsymbol{\beta}$. For each simulated patient, we generated a 10-dimensional vector of patient features, $X_1, \ldots, X_{10}$, drawn independently and uniformly distributed on $[-1, 1]$. Treatment $A$ was drawn from $\{-1, 1\}$ independently of the prognostic variables with $\mathbb{P}(A = 1) = 1/2$. The outcome variable $R$ was normally distributed with mean $Q_0 = 1 + 2X_1 + X_2 + 0.5X_3 + T_0(X, A)$ and standard deviation 1, where $T_0(X, A)$ was the interaction term between treatment and patient features. We examined two scenarios for the treatment-feature interaction term: Scenario 1 : $T_0(A, X) = (X_1 + X_2)A$ and Scenario 2 : $T_0(A, X) = 0.442(1 - X_1 - X_2)A$

Both scenarios 1 and 2 had linear decision boundaries determined by $X_1$ and $X_2$. For scenario 1, the true optimal rule was given by $\mathbb{1}(X_1 + X_2 > 0)$, while for scenario 2, it was $\mathbb{1}(1 - X_1 - X_2 > 0)$. OWL was implemented with a linear kernel. For Bayesian OWL, Gibbs sampling was used to draw from the posterior distributions of the parameters 500 times. The first 150 draws were discarded as "burn-in" and point estimates of $\boldsymbol{\beta}$ were computed by taking the mean of the draws from the posterior distribution. Throughout, we set the hyperparameter $\nu = 0.8$.

For each scenario, we varied the training dataset from 100 to 200, 400 and 800 and tested on 1000 patients. For each training set size, we conducted 200 simulation runs. We evaluated classification performance using the misclassification rate, $\frac{\text{Number of patients misclassified}}{\text{Total number of patients}}$. The simulation results are presented in Table 2.

As expected, the classification performance improved among all the ITR learning methods evaluated as the sample size increased. However, OWL consistently outperformed Bayesian OWL in all sample sizes and in both scenarios. We hypothesize that, with additional hyperparameter tuning, the performance of Bayesian OWL can be improved. Ordinarily, one would be hesitant to propose a method that is dominated by an existing method. However, the dominance of OWL is with respect to the misclassification rate. OWL, even with 800

| | | Scenario 1 | | |
|---|---|---|---|---|
| | | Bayesian OWL | Bayesian OWL | Bayesian OWL |
| $n$ | OWL | Normal Prior | Exponential Power Prior | Spike and Slab |
| 100 | 0.24 | 0.38 | 0.38 | 0.39 |
| 200 | 0.18 | 0.34 | 0.34 | 0.34 |
| 400 | 0.13 | 0.29 | 0.29 | 0.30 |
| 800 | 0.10 | 0.24 | 0.24 | 0.26 |
| | | Scenario 2 | | |
| | | Bayesian OWL | Bayesian OWL | Bayesian OWL |
| $n$ | OWL | Normal Prior | Exponential Power Prior | Spike and Slab |
| 100 | 0.22 | 0.38 | 0.38 | 0.39 |
| 200 | 0.15 | 0.34 | 0.34 | 0.34 |
| 400 | 0.13 | 0.31 | 0.31 | 0.30 |
| 800 | 0.10 | 0.25 | 0.25 | 0.22 |

Table 2: Misclassification rates for different methods and sample sizes.

samples in our simulation, has a 10%
misclassification rate, and there is no way to determine which of the 10% of the simulated patients
are likely misclassified (given a non-optimal treatment recommendation). In contrast, Bayesian OWL
yields the entire posterior distribution of the estimated optimal ITR and thereby allows for immediate
uncertainty quantification of individual-level treatment recommendations. In essence, Bayesian OWL
can inform us of which treatment recommendations it is less certain about whereas OWL cannot. We
demonstrate this in Section 4.2.

## 4.2 Treatment recommendation uncertainty

To highlight the utility of quantifying the uncertainty of individual-level treatment recommendations,
we trained Bayesian OWL model using simulated data as in Scenario 1 and the exponential power
prior. Then, we simulated another 1000 patients using the same generative approach, creating a fine
grid for the key variables $X_1$ and $X_2$, which are part of the true optimal rule. This allowed us to
estimate uncertainty across the domain of the true optimal ITR for all combinations of $X_1$ and $X_2$
within $[-1, 1]^2$.

Figure 1 shows how Bayesian OWL quantifies
uncertainty in its treatment recommendations.
The true optimal ITR assigns treatment $A = 1$ to
patients in the upper-right (where $X_1 + X_2 > 0$)
and treatment $A = -1$ to those in the lower-
left. Using the posterior predictive distribution
(Section 3.3), we calculated the uncertainty for
each treatment recommendation individually for
the 1000 patients in our test set. The heat map
in Figure 1 visualizes this uncertainty across
the ITR domain, with lighter colors (yellow and
light green) indicating higher certainty (close
to 1) and darker colors (purple and dark blue)
indicating lower certainty (close to 0). As ex-
pected, Bayesian OWL shows greater certainty
for patients with features far from the decision
boundary and less certainty for those near it. Fur-
thermore, in Figure 1, misclassified individuals

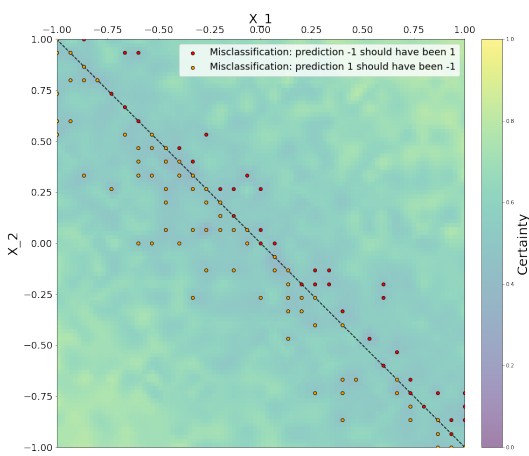

Figure 1: Heatmap of uncertainty quantification

are shown as red points for those incorrectly recommended treatment $-1$ instead of $1$ and orange
points for those incorrectly recommended treatment $1$ instead of $-1$. As expected, these misclassified
patients are near the decision boundary, particularly in regions where the model has the greatest
uncertainty (indicated by purple shading).

## 5 Discussion

We introduce a Bayesian formulation of OWL, the first Bayesian strategy for *directly* learning an
ITR from the classification perspective. Like OWL, our approach directly models the decision rule
without relying on conditional mean models. It also allows us to quantify uncertainty in treatment
recommendations at the *individual* level, unlike typical approaches focused on population-level
uncertainty. This insight can enhance clinical study design by identifying patient types where
treatment recommendations are confident and those needing further sampling.

Our work has some limitations. We only consider linear rules, but nonlinear rules may sometimes be
more appropriate or offer significant clinical improvements. Henao et al. [2014] proposed a Bayesian
SVM for nonlinear decision boundaries, which could be adapted for this purpose. Additionally, we
have not fully explored variable selection as in Song et al. [2015], limiting its use in high-dimensional
settings.

The classification approach for optimal ITRs is powerful, but it struggles with inference and uncer-
tainty quantification. Bayesian OWL addresses this by combining the strengths of direct learning
with a probabilistic framework, expanding inferential potential and improving precision medicine
evidence.

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
