# OpenReview forum: "Bayesian Outcome Weighted Learning"
_NeurIPS.cc/2024/Workshop/BDU — NeurIPS BDU Workshop 2024 Poster_

### Official Review · Reviewer_qYuV · 2024-09-19

**Rating:** 5
**Confidence:** 3

**Review:**

**Summary**: This work presents a Bayesian approach to the outcome-weighted learning (OWL), a classification-based framework for estimating optimal individualized treatment regimes (ITR). The motivation behind this approach is to incorporate uncertainty in the estimation and prediction of optimal ITRs, which is crucial for decision-making and applications in personalized medicine.

**Evaluation**
- **Quality (6/10)**: The proposal is of acceptable quality, but it needs refinement and expansion in key areas to elevate it to a higher scientific standard.
- **Clarity (6/10)**: Overall, the manuscript conveys its main ideas within the four-page limit. Yet, certain key points could benefit from further elaboration. For example, the discussion of the pseudo-posterior is somewhat lacking, particularly in clarifying that it is not strictly Bayesian, as it isn't derived from an actual data-likelihood. Additionally, some of the notation changes add confusion, such as using $r_i$ in the product $r_ix_i^\top\beta$ in section 2.3, but switching to $a_i$ in $a_ix_i^\top\beta$ at the start of section 3.
- **Originality (5/10)**:  Although the proposal of a pseudo-Bayesian OWL method for estimating ITR is somewhat original, it seems to me that the approach largely builds on and adapts concepts from Bayesian SVMs. I would have liked to see more methodological innovation addressing the unique challenges of learning optimal ITR specifically.
- **Significance (5/10)**: I appreciate the motivation to introduce Bayesian-like and probabilistic modeling tools to the challenge of estimating ITR with proper uncertainty quantification. In my view, aside from the contributions of T. Murray, S. Murphy, and a few others, there is indeed a notable gap in the literature on this topic, primarily due to objections to likelihood-based inference in modern semiparametric causal inference. This work makes progress by proposing a pseudo-Bayesian approach. However, it falls short in some areas. For instance, it relies on data from RCTs with binary treatments, which are often scarce, small in size, and prone to selection bias. Extending the approach to observational settings, where $\rho$ is neither known nor fixed but must be estimated from a sufficient set of confounders, would increase its applicability to real-world healthcare scenarios

**Pros**
- The problem is well motivated.

**Cons**
- The approach imposes significant constraints, such as focusing on linear optimal ITR, low-dimensional settings, and experimental data with randomized binary treatments, which limit its broader applicability.
- Lacks a clear benchmark comparison highlighting its strengths and weaknesses relative to Bayesian $Q$-learning for ITR.

---

### Official Review · Reviewer_fwVr · 2024-09-26
**it is unclear to me whether this work focuses on developing an approach or an application**

**Rating:** 5
**Confidence:** 3

**Review:**

**Summary**

This paper introduces a Bayesian approach to Outcome Weighted Learning (OWL) for learning optimal individualized treatment regimes (ITRs) with uncertainty quantification. The key contribution is the transformation of the OWL framework into a Bayesian probabilistic model, which allows for inference and individual-level uncertainty quantification.

**Strengths**
1. The manuscript introduces a novel Bayesian approach to OWL, which addresses key limitations in the field, such as the lack of uncertainty quantification in traditional machine learning methods.
2. The proposed method directly learns optimal ITRs without needing to estimate conditional mean models, which provides an innovative contribution to Bayesian machine learning in healthcare applications.
3. The simulation study and discussion highlight the potential of this approach in precision medicine, with practical implications for improving treatment decisions.

**Weakness**
1. An abstract is absent in the manuscript.

2. While the paper claims to present a Bayesian learning method, it frequently shifts focus toward a specific application within individualized treatment regimes. This introduces terminology and concepts, such as biomarkers, prognostic information vectors, and cohorts, which may be unfamiliar to readers who are not experts in healthcare or precision medicine. These domain-specific terms can obscure the core contributions related to Bayesian optimization. The manuscript would benefit from a clearer focus on whether the work is primarily about developing a Bayesian learning framework or applying it to healthcare, with adequate explanations of these concepts for non-experts.

3. Figure 1 could be improved in terms of readability. The current use of lighter and darker colors is not easily distinguishable. Moreover, the labels, numbers, and markers are too small to read comfortably.

**Questions**
1. Can the authors clarify the specific aspects of the Bayesian approach that enable uncertainty quantification at the individual level?
2. How does the proposed method generalize beyond the healthcare domain?

---

### Decision · Program_Chairs · 2024-10-09

**Decision:**

Accept (Poster)

**Comment:**

This paper is borderline. The reviewers' actual text in general reads more positive than the scores, and one of the reviewers' own breakdown lists higher numbers. Reviewers in particular are concerned about lack of comparisons, but also note that the method is reasonably novel, which might be contributing to this. Overall my impression is this paper lands slightly above the decision boundary, and I recommend accepting this.